# Inflammatory Molecules Associated with Ultraviolet Radiation-Mediated Skin Aging

**DOI:** 10.3390/ijms22083974

**Published:** 2021-04-12

**Authors:** Tuba M. Ansary, Md. Razib Hossain, Koji Kamiya, Mayumi Komine, Mamitaro Ohtsuki

**Affiliations:** Department of Dermatology, Faculty of Medicine, Jichi Medical University, Tochigi 329-0498, Japan; tuba2020@jichi.ac.jp (T.M.A.); razib@jichi.ac.jp (M.R.H.); m01023kk@jichi.ac.jp (K.K.); mamitaro@jichi.ac.jp (M.O.)

**Keywords:** inflammation, ultraviolet radiation (UVR), skin aging

## Abstract

Skin is the largest and most complex organ in the human body comprised of multiple layers with different types of cells. Different kinds of environmental stressors, for example, ultraviolet radiation (UVR), temperature, air pollutants, smoking, and diet, accelerate skin aging by stimulating inflammatory molecules. Skin aging caused by UVR is characterized by loss of elasticity, fine lines, wrinkles, reduced epidermal and dermal components, increased epidermal permeability, delayed wound healing, and approximately 90% of skin aging. These external factors can cause aging through reactive oxygen species (ROS)-mediated inflammation, as well as aged skin is a source of circulatory inflammatory molecules which accelerate skin aging and cause aging-related diseases. This review article focuses on the inflammatory pathways associated with UVR-mediated skin aging.

## 1. Introduction

Human skin is the largest organ in the body and protects our internal organs from the external world and is thus exposed to different types of hazardous environmental stimuli. Skin aging is one of the most visible signs of human aging dependent on chronological phenomena (intrinsic aging) and external factors (extrinsic aging or photoaging). Intrinsic aging is a normal physiological process characterized by decreased cell proliferation in basal layers and accumulation of senescent cells in epidermis and dermis resulting in skin dryness, thinning, fine wrinkles, itching [1], and susceptibility to many skin disorders, such as infections, autoimmune disorders, and malignancy [2]. External factors, including pollutants, smoking, diet, temperature, and especially ultraviolet and infrared radiation have larger influence on aggravation of inflammation and aging phenotypes such as wrinkles, irregular pigmentation, skin dryness, and decreased dermal and epidermal thickness [3]. 

The ultraviolet radiation (UVR) is emitted naturally from the sun and artificial sources and can cause severe damage to the skin, referred to as sun burn. The major artificial sources of UVR are mercury vapor lamps, water-cooled lamps, and air-cooled lamps mainly used for diagnosis and treatment purposes. Chronic exposure to UVR with less dosage causes sun-tan and accelerates skin aging, called photoaging. UVR from sunlight can be classified into three types by their wavelength: UVA (320–400 nm), UVB (280–320 nm), and UVC (100–280 nm) [4]. Among them, almost all UVC and some UVB are absorbed by the ozone layer and do not have any impact on our skin [5,6]. The rest of the UVB can penetrate the skin epidermis and cause erythema (sunburn), while UVA can invade the dermis and is about 98% responsible for major skin aging [7] (Figure 1).

UVB absorbed by the epidermal cells causes DNA damage, increases oxidative stress, reactive oxygen species (ROS), and leads to premature aging [8,9]. UVA, on the contrary, has a higher wavelength that can cause indirect DNA damage along with collagen and elastin fiber degradation through oxidative stress pathways [4]. Altogether, chronic exposure to UVR results in an increase in nicotinamide adenine dinucleotide phosphate (NADPH) oxidase and generates ROS, which elevates inflammation, cytokines, chemokines, and skin aging [10]. Chronic and persistent inflammation caused by UVR can weaken skin defense mechanisms and degrade collagen and elastin fibers, and ultimately lead to premature aging. The purpose of this review is to discuss the inflammatory molecules associated with UVR-mediated skin aging. In this review, our main focus is to elaborate the effects of UVR on epidermal keratinocytes and dermal fibroblast in the skin aging process.

## 2. Inflammatory Response Caused by UVR

It is well-understood that the acute response of skin to UVR is inflammation, such as erythema and edema, and DNA and mitochondrial damage caused by ROS [11]. The major ROS species involved in this process are superoxide radical anion (O_2_^−^), hydroxyl radical (OH), hydrogen peroxide (H_2_O_2_), and singlet oxygen species (O_2_) [12]. It has been reported that UVB (290–320 nm) exposure caused a significant increase in ROS production in human epidermal keratinocytes (HaCaT) cells along with a decrease in cell viability [13]. ROS is a byproduct of regular oxygen metabolism and is involved in many physiological functions, such as cell signaling, enabling defense mechanisms against pathogens, and cell proliferation. However, UVR exposure to skin can produce increased amounts of ROS, which causes an imbalance between ROS production and antioxidant defense mechanisms, resulting in oxidative stress [10,14]. In a vicious cycle, this oxidative stress can increase ROS production, and can initiate both inflammation and pro-inflammatory cytokine activation, such as interleukin-2 (IL-2), interleukin-6 (IL-6), and tumor necrosis factor α (TNF-α), involving multiple pathways including nuclear factor kappa light chain enhancer of activated B (NF-kB), hypoxia-inducible factor 1-alpha (HIF-1a), nuclear factor erythroid 2-related factor 2 (Nrf-2), and activator protein 1 (AP-1) [15,16]. UVR-related inflammation is also associated with klotho deficiency. Klotho is a transmembrane protein which is also known as an anti-aging hormone protecting against different stressors [17]. Several studies have confirmed that klotho’s function is mediated by the NF-κB pathway. UV irradiation caused a decrease in klotho mRNA and protein expression along with increased pro-inflammatory cytokine expression, such as interleukin-1 β (IL-1β), interleukin-6 (IL-6), and TNF-α in HaCaT cells. Moreover, klotho overexpression in human keratinocytes decreased the UV-induced cell damage and inflammation by inhibiting NF-κB nuclear translocation and it also decreased H_2_O_2_-induced inflammation by inhibiting Toll-like Receptor 4 expression [18,19]. Sirtuins, a nicotinamide adenine dinucleotide (NAD(+))-dependent histone deacetylase, is also gaining attention for its ability to increase lifespan as it can delay cellular senescence and promote DNA damage repair. It has been shown that SIRT1, a member of the Sirtuins family, displays an anti-inflammatory effect by inhibiting NF-κB signaling. UVR can also cause chronic inflammation by downregulating SIRT1 expression in human keratinocytes [20,21] (Figure 2).

### 2.1. Major Inflammatory Responses in Epidermis upon UV Exposure

The outermost skin layer is the epidermis, and it is constantly renewing and differentiating. It also works as a barrier against the outer world and is most directly affected by the surrounding environment, mainly UVR. The epidermis mainly consists of four types of cells: predominantly keratinocytes (~90%), melanocytes, Langerhans’ Cells and Merkel Cells [22]. Keratinocytes form a water barrier by means of the stratum corneum (SC), which is generated in the epidermal basal layer, and the tight junctions form a barrier in the stratum granulosum [23]. Almost all UVB is absorbed by the SC, the outermost layer of the epidermis. Several papers have investigated the effects of UVR in epidermis [24,25,26]. The major SC damage caused by UV exposure includes rough and dehydrated texture, reduced desquamation and barrier function, and detrimental effects on cell cohesion [27]. The chronic UV-irradiated epidermis is characterized by thinning of epidermis, fine wrinkles, dryness, and disrupted epidermal barrier function [26,28]. In vitro studies demonstrated an increase in epidermal thickness upon UV irradiation in human samples, whereas in a clinical study, a gradual decrease in epidermal thickness in areas exposed to minimal sunlight has been reported [29,30,31,32]. This discrepancy depends on the chronicity of UV exposure. Acute stimulation with UV enhances keratinocyte proliferation though activating epidermal growth factor receptor (EGFR), while chronic exposure to sunlight accelerates aging processes, which render epidermis thinner with flattening of rete ridges (Figure 3). On the other hand, non-sun-exposed areas of aged people show comparable thickness to those of young people [33]. 

It has been demonstrated that UVB exposure disrupted the epidermal barrier function in male hairless Balb/c mice in a dose-dependent manner [34,35], and skin barrier disruption can lead to acute inflammatory response or exacerbation of chronic inflammatory skin diseases, such as atopic dermatitis [36]. Transepidermal water loss (TEWL) is commonly known as a parameter for measuring skin barrier disruption and many reports have confirmed that different UV doses can cause increase of TEWL in murine and human samples [37,38,39,40]. In epidermis, there are several inflammatory signaling pathways connected to different surface receptors, such as the EGFR [41], transforming growth factor receptors (TGFR) [42,43], toll-like receptors (TLRs) [44], IL-1 receptor, and TNF receptor (TNFR) [45,46,47]. The dominant source of cytokines in epidermis is keratinocytes, and major cytokines secreted from keratinocytes upon UVR irradiation are Interleukins (IL-1, IL-3, IL-6, IL-8, IL-33), colony-stimulating factors (GM-CSF, M-CSF, G-CSF), and transforming growth factor α (TGF-α), transforming growth factor β (TGF- β), TNF-α, high-mobility group box 1 (HMGB1), and platelet-derived growth factor (PDGF) [48,49,50]. UVR can activate signaling directly by ROS production or indirectly by DNA or mitochondrial damage and then trigger inflammation. Previous studies regarding UVR-mediated inflammation are summarized in Table 1.

EGFR signaling plays an important role in keratinocyte proliferation, differentiation, cell adhesion, migration, and survival. However, the role of EGFR on inflammatory response upon UVR is very complex. One report demonstrated that UV-irradiated mice and keratinocytes treated with EGFR inhibitor showed suppressed inflammatory responses, such as decreased immune cell infiltration and decreased levels of inflammatory cytokines (TNF-α, IL-8, IL-1a, Cyclooxygenases (COX2)) [77]. Results also showed that EGFR is partially responsible for p38 mitogen-activated protein kinase (MAPK)-activated inflammatory response [77]. It is well-known that p38 MAPK signaling is involved in varied pro-inflammatory cytokine production, leading to various skin pathogenesis, including photoaging. UV-induced COX-2 expression is mostly dependent on the p38 MAPK signaling pathway [78] and regulates prostaglandin 2 (PGE2) secretion, which are major pro-inflammatory cytokines responsible for immune cell infiltration, swelling, and edema [79]. However, some recent reports have shown that prolonged EGFR inhibitor treatment can lead to premature skin aging phenotypes by ROS-induced oxidative stress, or cellular senescence induced by cell cycle arrest [80]. Therefore, the role of EGFR on UVR-mediated skin aging is very complex and yet to be elucidated. 

NF-κB is one of the major mediators of cellular inflammatory processes and it is well-established that UV irradiation can increase NF-κB transcriptional activity, resulting in chronic inflammatory signal [81]. Human keratinocytes irradiated with UV showed increased expression of inflammatory cytokines IL-1β, IL-6, IL-8, and TNF-α through the NF-κB pathway [82,83]. TLRs are expressed in epidermal keratinocytes and Langerhans cells and are crucial in pathogen identification and immune responses. It has been demonstrated TLRs have important function in UV-mediated inflammation through its downstream signaling pathway involving NF-κB. Specifically, UV-damaged keratinocytes secrete noncoding RNA that can activate TLR3 and induce inflammatory responses, such as TNF-α and IL-6 [84]. Skin epidermis has predominant expression of TNFR, and TNF-α can activate various inflammatory pathways through NF-κB and MAPK. It has been reported that UV irradiation significantly increased both soluble and full-length TNF-α in epidermal keratinocytes [85]. Although epidermis is constantly renewing and cell apoptosis is an essential factor for the epidermis homeostasis, it is also important to note that impaired, premature, or excessive apoptosis can lead to epidermal homeostasis dysregulation and promote aging phenotypes, such as sunburn cells. It has been proven that TNF-α can cause keratinocyte apoptosis by the UV-induced TNFR-1 or p55 receptor pathway [86,87]. 

The expression of TGF- β is comparatively lower in human epidermis, which is mostly expressed in epidermal basal layers and responsible for epithelial homeostasis, wound healing, and anti-inflammatory response [88]. Chronic UV irradiation can decrease the TGF-β 2 synthesis by reducing TGF-β type II receptor (TβRII) mRNA expression [89,90].

### 2.2. Major Inflammatory Responses in Dermis upon UV Exposure 

The dermis is mainly composed of collagen, elastic fibers, nerves, blood vessels, hair follicles, and glands, where the major component of dermis is collagen [91]. The major cell types in dermis are fibroblasts, vascular smooth muscle cell, macrophages, adipocytes, mast cells, schwann cells, and follicular stem cells [91,92]. Fibroblasts provide dermis with collagen-rich extracellular matrix (ECM) and immune cell infiltration is maintained by blood vessels and lymphatic vessels [93,94]. Photoaged dermis is mostly characterized by thin dermis, decreased collagen content, disorganized and fragmented collagen fibers, elastic fiber degradation, and severe dermal connective tissue damage, which results in most visible signs of skin aging, such as wrinkle formation, fragile atrophic skin, delayed wound healing, and sagging [95]. There are many in-vivo and in-vitro studies that have explored the effects of UVR on dermis (Table 1). 

Damaged collagen fibrils and elastin fibers are the most evident component of UVR-mediated aged dermis, mainly caused by the matrix-degrading metalloproteinases (MMPs) synthesis through MAP kinase signaling [14]. MMPs are a family of ubiquitous endopeptidases and participate in inflammatory processes by regulating chemokine activity [96,97]. It has been reported that MMPs are responsible for collagen degradation, mainly through collagenase-1 (MMP-1), stromelysin-1 (MMP-3), and gelatinase B (MMP-9), which directly and fully degrade collagen [98]. One report demonstrated increased expression of MMP-1, MMP-2, MMP-3, MMP-9, MMP-11, MMP-17, and MMP-27 in a photoaged human forearm correlated with reduced type I procollagen expression [99]. Type I collagen is the most abundant protein in the ECM, and collagen fibrils are synthesized by the procollagens (collagen precursor molecules) through a series of reactions [100]. Another report showed that MMP-1 expression was increased in human UV-irradiated skin along with collagen degradation [101].

UVR can also initiate collagen degradation by reducing procollagen synthesis, and it has been reported that UVR can reduce procollagen synthesis by downregulating the TGF-β/Smad signaling pathway [102]. TGF-β initiates signaling through binding with cell surface receptors (TGF-β type I and type II receptors), then transcription of TGF-β-associated genes by phosphorylating Smad2/Smad3 complex. It has been reported that TGF-β can activate promoter activity of Type I collagen gene (COL1A2) through the Smad signaling pathway [103]. One report showed that UV irradiation inhibits the Smad2/Smad3 nuclear translocation, resulting in decreased TGF-β type II receptor transcription and protein synthesis and procollagen type I expression in hairless mice and in human dermal fibroblasts [76,102,104] (Figure 4).

Various stressors including UV radiation can activate DNA damage response that can initiate cell cycle arrest through the p53/p21 pathway involving the P38/MAPK cascade and NF-κB [105] pathway. It has been demonstrated that UV-exposed human skin has significantly high accumulation of senescent cells [106]. Accumulation of senescent fibroblasts can accelerate skin aging by secreting senescence-associated secretory phenotype (SASP) factors, including IL-1α, IL-1β, IL-6, IL-8, and MMPs. SASP factors secreted from senescent fibroblasts are responsible for chronic inflammation as well as ECM degradation, resulting in photoaging [107]. Figure 5 explains the major signaling pathways involved in UV-mediated photoaging.

## 3. Conclusions

This review summarized the association between the inflammatory molecules and skin aging mediated by UVR. Natural aging can be accelerated by external factors, mainly UVR, which causes direct damage to the DNA or indirectly by producing radicals. In skin, epidermal keratinocytes are the major source of cytokines production and dermal fibroblasts are the major source of MMPs, and it has been demonstrated that DNA damage leads to release of inflammatory molecules from epidermal keratinocytes (IL-1, IL-3, IL-6, IL-8, GM-CSF, M-CSF, G-CSF, TGF-α, TGF-β, TNF-α, and PDGF) and from dermal fibroblasts (MMP-1, MMP-2, MMP-3, MMP-9, MMP-11, MMP-17, and MMP-27) [48]. Chronic inflammation can exacerbate aging and result in secretion of SASP factors in dermal fibroblasts and melanocytes [108,109]. Chronic SASP factor secretion is responsible for many age-related pathologies, for example atherosclerosis, type 2 diabetes, obesity, cardiovascular disease, sarcopenia, neurodegenerative diseases, and Alzheimer’s disease [110,111,112,113]. Since it has been established that inflammatory responses after UV exposure can accelerate the aging process as well as age-associated pathophysiology, new approaches targeting these inflammatory molecules need to be developed.

## Figures and Tables

**Figure 1 ijms-22-03974-f001:**
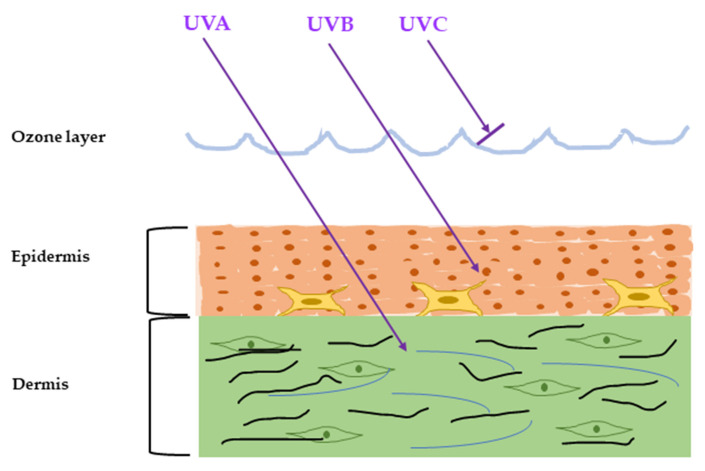
Penetration of the solar ultraviolet radiation (UVR) into the skin. According to the wavelength, the UVR is classified into three categories: UVA, UVB, and UVC. UVC is blocked by the ozone layer, UVB can penetrate into the epidermis, and UVA can penetrate up to the dermis.

**Figure 2 ijms-22-03974-f002:**
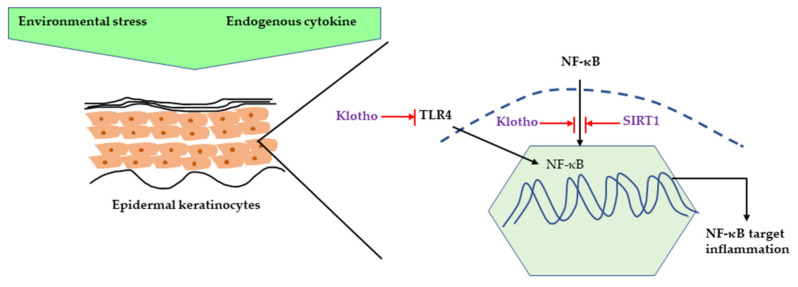
Inhibition of NF-κB-induced inflammation by klotho and SIRT1. NF-κB can be activated by different environmental stimuli as well as endogenous cytokines, such as TNF-α and IL-1. Klotho can inhibit the NF-κB pathway by preventing translocation of NF-κB or inhibiting TLR4-mediated NF-κB activation. SIRT1 prevents the NF-κB pathway by directly inhibiting NF-κB deacetylating NF-κB subunit. NF-κB, nuclear factor kappa light chain enhancer of activated B; TNF-α, Tumor necrosis factor; IL-1, Interleukin 1; TLR4, Toll-like receptor 4, SIRT1, Sirtuin 1.

**Figure 3 ijms-22-03974-f003:**
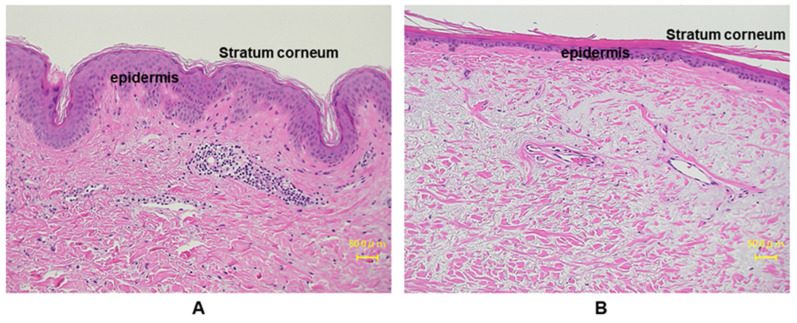
Histological comparison between young and aged photo-exposed human skin. Skin specimens from (**A**) young: 12 years, and (**B**) aged: 77 years, are stained with hemotoxylin and eosin (HE) staining. Aged human skin epidermis is thinner compared to the young skin, as well as the aged dermis displays increased amounts of degenerated elastic fibers.

**Figure 4 ijms-22-03974-f004:**
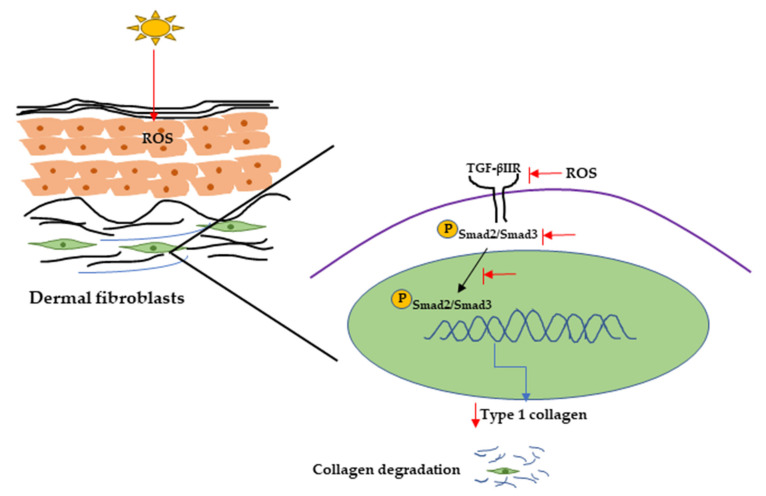
UVR-induced collagen degradation through the TGF-β pathway. UV radiation can cause a reduction in TGF-β type 2 receptor expression by producing ROS. It can also inhibit Smad2/Smad3 phosphorylation and nuclear translocation, resulting in decreasing Type 1 collagen expression and collagen degradation. UV, Ultraviolet; TGF-β, transforming growth factor; ROS, reactive oxygen species.

**Figure 5 ijms-22-03974-f005:**
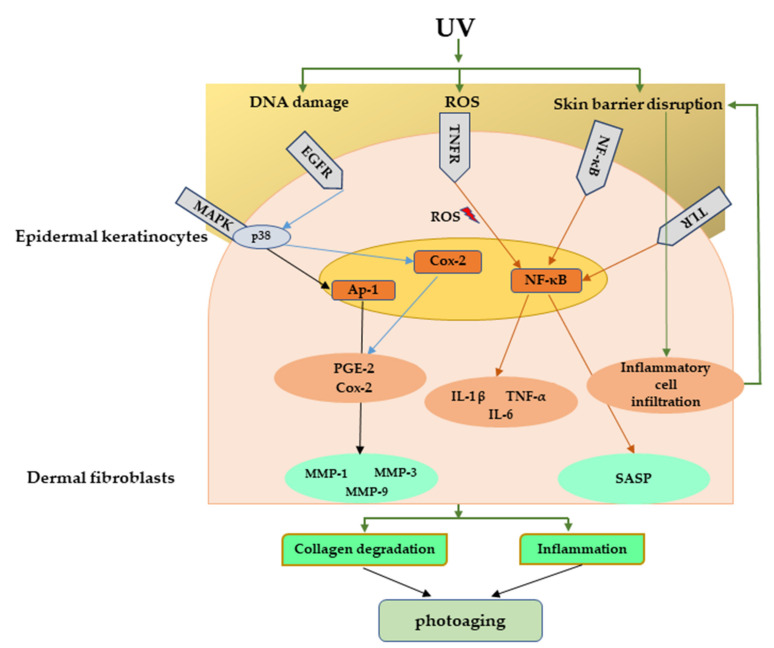
Major signaling pathways involved in UV-mediated photoaging. UV radiation can cause direct damage to the DNA, produce ROS, and disrupt skin barrier function. These can further activate many inflammatory receptors, which can lead to photoaging through inflammation and collagen degradation. UV, Ultraviolet; ROS, reactive oxygen species.

**Table 1 ijms-22-03974-t001:** Ultraviolet radiation (UVR)-mediated inflammation.

Study Type	Study Subject	UV Dose	Inflammatory Cytokine	Aging Phenotype	Reference
In-vivo	DBA/2 mice	180 mJ/cm^2^	TNF-α	Increase epidermal thickness, neutrophil infiltration	[47]
In-vivo	HR-1 hairless mice	100 mJ/cm^2^	TNF-α, COX-2, iNOS, IL-6, IL-1β	Skin wrinkle formation, increase epidermal thickness, collagen degradation, mast cell infiltration	[51]
In-vivo	HR-1 hairless mice	100 mJ/cm^2^	TNF-α, IL-6, IL-1β	Skin wrinkle formation, increase epidermal thickness, collagen degradation, trans epidermal water loss (TEWL) of dorsal skin	[52]
In-vivo	Chinese Kun Ming mice	100–400 mJ/cm^2^	IL-1β, IL-6, TNF-α, COX-2, PGE2, MMP-1, MMP-3	Coarse wrinkles, erythema, edema, thickening, leathery appearance, epidermal hyperplasia, reduced collagen fibers	[53]
In-vivo	SKH-1 hairless mice	100 mJ/cm^2^	TNF-α, MMP-13, IL-1 β, IL-6	Increase epidermal thickness, collagen degradation	[54]
In-vivo	hairless mice (HRS/J)	0.384 mW/ cm^2^	TNF-α, IL-6, IL-1β	Increase epidermal thickness, collagen degradation, mast cell infiltration	[55]
In-vivo	hairless mice	312 nm and 790 µW/cm^2^ intensity	IL-6, IL-12, TNF-α	Skin wrinkle formation, increase epidermal thickness, collagen degradation, collagen and elastic fibers’ (elastin) degradation	[39]
In-vivo	ICR mice	UVA: 20.81 J/cm^2^, UVB: 0.47 J/cm^2^	MMP-1, MMP-3, TNF-α	Thicker scarf-skin, and a disruption of the skin tissue in dermis	[56]
In-vitro	HaCaT and HDFs	200 mJ/cm^2^	TNF-α, IL-1β	Increase senescence-associated β -galactosidase activity, increase ROS production	[57]
In-vivo	Wister rat	280–400 nm	IL-6, IL-1β, TNF-α	Increase epidermal thickness, disrupted stratum corneum, abnormal hair follicles and loss of histological architecture, uneven sebaceous glands dermis, reduced skin collagen and elastic fibers	[58]
In-vivo and In-vitro	HaCaT	20 mJ/cm^2^	COX-2, MMP-1, FGF, TNF-α, IL-6, and decreased TGF-β1	Increase ROS production, DNA oxidative damage, reduce procollagen type I	[59]
	Kunming mice	150–300 mJ/cm^2^	Not investigated	Increase epidermal and dermis thickness, infiltrationof inflammatory cells, decrease collagen fibers	
In-vivo	hairless BALB/c	55 mJ/cm^2^	TNF-α, IL-1β, IL-6	Increase epidermal and dermis thickness, skin erythema, dry, thickening, sagging, coarse wrinkles, and reduced skin collagen type I	[60]
In-vitro and ex-vivo	NHDFs	144 mJ/cm^2^	TNF-*α*, IL-6, iNOS, COX-2	Increase ROS production	[61]
	Reconstructed human skin (Keraskin™ FT)	100 mJ/cm^2^	Not investigated	Wrinkle formation, disruption and decomposition of collagen fibers in skin tissues exposed to UVB	
In-vivo and In-vitro	HaCaT	75 mJ/cm^2^	TNF-*α*, IL-6, MMP-1	Not investigated	[62]
	hairless mice	100 mJ/cm^2^	MMP-9, MMP-13	Increase epidermal and dermis thickness	
In-vivo	Human	25–50 J/ cm^2^	IL-6	Not investigated	[63]
In-vivo	hairless mice	45–210 mJ/cm^2^	MMP-2, 9, 12 and 13, TNF-α, IL-1β	Increase epidermal thickness, decrease Type I collagen, deep wrinkle formation	[64]
In-vivo	Human	70 and 90 mJ/cm^2^	TNF-α, MMP-1	Decrease Type I procollagen	[65]
In-vivo	Kunming mice	1 to 4 MED, 1 MED = 70 mJ/cm^2^	IL-1β, IL-6, IL-10, TNF- α, MMP-1, MMP-3	Reduce skin elasticity, coarse wrinkle formation, erythema, increase epidermal thickness, decrease collagen content	[66]
In-vivo and In-vitro	HaCaT cell	10 mJ/cm^2^	IL-1α, IL-1β, IL-6, TNF-α, MMP-2/9	Increase ROS production	[67]
	BALB/c mice	180 mJ/cm^2^	Not Investigated	Increase epidermal thickness, infiltration of leucocyte	
In-vivo	SKH-1 hairless mice	30 to 120 mJ/cm^2^	MMP-1, 2, 9, COX2, IL-1β, IL-6	Increase epidermal and dermal thickness, thick and deep wrinkle formation, decrease collagen content	[68]
In-vivo and In-vitro	HaCaT cell	50 J/m^2^	COX-2 IL-1β, IL-6, TNF-α, MMP1, MMP2, MMP9	Increase cellular senescence	[69]
	BALB/c athymic nude mice	60 to 120 mJ/cm^2^	COX-2, IL-1β	Increase epidermal thickness, wrinkle formation	
In-vivo	Kunming mice	1 to 4 MED, 1 MED = 70 mJ/cm^2^	IL-1β, IL-6, IL-10, TNF-α	Deep wrinkles, erythema, edema, and skin burn, increase epidermal thickness, degrade dermal collagen fibers, suppress antioxidant enzymes	[70]
In-vivo	Kunming mice	75 to 300 mJ/cm^2^	MMP-1, MMP-3, IL-6, TNF-α, IL-1	Edema, erythema, thickening and coarse wrinkles, epidermal hyperplasia, disorganized collagen fibers	[71]
In-vitro	HaCaT cell	15 mJ/cm^2^	COX-2, TNF-α, IL-1β	Increase ROS production, suppress antioxidant enzymes	[72]
In-vivo and In-vitro	HaCaT cell	12.5 mJ/cm^2^	TNF-α, IL-1 α, MMP-1, COX-2	Not investigated	[73]
	SKH-1 hairless mice	36–122 mJ/cm^2^	MMP-1	Increase epidermal and dermal thickness, decrease Type I collagen	
In-vivo	KM mice	1–4 MED, 1 MED = 100 mJ/cm^2^	IL-10, IL-6, TNF-α, MMP-1, MMP-3	Severe wrinkles, increase epidermal thickness, decease antioxidant enzymes	[74]
In-vivo and In-vitro	NHDFs	90 mJ/cm^2^	MMP-1, IL-1 α, IL-1β, IL-6, TNF-α	Increase ROS production	[75]
	BALB/c mice	90 mJ/cm^2^	IL-8	Wrinkle formation, epidermal hyperplasia	
In-vivo	SKH-1 hairless mice	100–200 mJ/cm^2^	MMP-1, MMP-9, decreased TGF-β1	Increase epidermal thickness, reduce procollagen type I	[76]
In-vivo	SKH-1 hairless mice	280–320 nm	Reduced klotho	Hyper-thickened epidermis	[18]

HaCaT, Human epidermal keratinocytes; HDFs, Human dermal fibroblasts; NHDFs, Normal human dermal fibroblasts, HSF2; Human skin fibroblasts; MED, minimal erythemal dose; TNF-α, tumor necrosis factor-a; COX-2, cyclooxygenase-2; iNOS, Inducible nitric oxide synthase; IL, Interleukin; PGE2, Prostaglandin E_2_; MMP, Matrix metalloproteinase; FGF, Fibroblast growth factors; TGF-β1, Transforming growth factor beta 1; ROS, reactive oxygen species.

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
