# Peer review of "Inflammatory Molecules Associated with Ultraviolet Radiation-Mediated Skin Aging"

_ijms, 2021, doi:10.3390/ijms22083974_

Round 1

Reviewer 1 Report

This is a nice review looking at the inflammatory responses of skin to UVR. I enjoyed reading it very much. I would have enjoyed it much more if the manuscript had first been checked by a  native english speaker and I would request that the authors do this for resubmission. Other than the english, I have only the following minor comments:

1) abstract line 13 -'decreased permeability' - are you sure this is what you mean? The concensus is that UVR increases permeability because the barrier function is weakened.

2) Line 39 - in the discussion of UVB, please change 'pass through the epidermis' to 'penetrates the epidermis'

3) line 97 - Lipid production is not the only mechanism by which the epidermis forms a water barrier. I would suggest amending this to  'Keratinocytes form a water barrier by means of the stratum corneum and tight junctions in the granular layer' or something similar

4) line 109 - the figure and the text do not match

'On the other hand, non-sun-exposed areas of aged people shows compara- 
ble thickness to that of young people. [31]. (Fig. 3). ' - Figure 3 shows photoaged skin, not aged skin'. I think the reference to figure 3 is in the wrong place. Please amend for clarity.

Author Response

Response to Reviewer 1 Comments

This is a nice review looking at the inflammatory responses of skin to UVR. I enjoyed reading it very much. I would have enjoyed it much more if the manuscript had first been checked by a native English speaker and I would request that the authors do this for resubmission. Other than the English, I have only the following minor comments.

We thank the reviewer for the thoughtful review and crucial comments which have helped guide the revision of the manuscript. Our replies are as follows:

Point 1: abstract line 13 -'decreased permeability' - are you sure this is what you mean? The concensus is that UVR increases permeability because the barrier function is weakened.

Response 1: We apologize for this unintentional mistake. We have made a correction in abstract (page 1, line 14).

“Skin aging caused by UVR is characterized by loss of elasticity, fine lines, wrinkles, reduced epidermal and dermal components, increased epidermal permeability, delayed wound healing and approximately 90% of skin aging.”

Point 2: Line 39 - in the discussion of UVB, please change 'pass through the epidermis' to 'penetrates the epidermis.

Response 2: We would like to appreciate the reviewer`s suggestion regarding the correct word choice. We have modified the sentence according to the suggestion (page 2, lines 43).

“The rest of the UVB can penetrate the skin epidermis and cause erythema (sunburn), while UVA can invade the dermis and is about 98% responsible for major skin aging [7].”

Point 3: line 97 - Lipid production is not the only mechanism by which the epidermis forms a water barrier. I would suggest amending this to ‘Keratinocytes form a water barrier by means of the stratum corneum and tight junctions in the granular layer' or something similar. 

Response 3: We would like to appreciate the reviewer`s suggestion regarding how the epidermis forms a water barrier. We have modified that sentence according to the reviewer’s comment (page 3, lines 20-121).

“Keratinocytes form a water barrier by means of the stratum corneum (SC), which is generated in the epidermal basal layer and the tight junctions, form a barrier in stratum granulosum [23].”

Point 4: line 109 - the figure and the text do not match.

'On the other hand, non-sun-exposed areas of aged people shows comparable thickness to that of young people. [31]. (Fig. 3). ' - Figure 3 shows photoaged skin, not aged skin'. I think the reference to figure 3 is in the wrong place. Please amend for clarity

Response 3: We would like to apologize for this mistake. We have amended the sentence (page 4, line 130).

“Acute stimulation with UV enhances keratinocyte proliferation though activating epidermal growth factor receptor (EGFR), while chronic exposure to sunlight accelerates aging processes, which render epidermis thinner with flattening of rete ridges (Fig. 3).”

Response to Reviewer 2 Comments

This review summarizes skin aging caused by UVR and subsequent ROS mediated inflammation and associated circulatory inflammatory molecules and their inflammatory pathways. The topic is interesting and the review provides a relevant compilation of studies and important findings, in particular in the form of a comprehensive table. It is my opinion that the review could be published after minor revisions.

In general, the language could be improved (a linguistic review is recommended). Likewise, the figures could be edited in a more compelling manner, including larger font size of the labels in the figures.

We thank the reviewer for the thoughtful review and important comments which have helped guide the revision of the manuscript. According to the reviewer’s suggestions we have completed a thorough English corrections and also increase the font size in figure legends. Our replies are as follows:

Point 1: The "table" should be referred to as Table 1.

Response 1: We thank the reviewer`s comment regarding how Table should be referred.  We have also modified accordingly (page 4, line 159, page 5, 197).

Point 2:  Lines 56-57: ROS cause inflammation following acute exposure to UVR. Please include the major/most relevant ROS species in this instance.

Response 2: We thank the reviewer for this comment regarding the major ROS species involved in UVR mediated inflammation. In the revised manuscript, we have added these information (page 2, line 74-75). 

“It is well understood that the acute response of skin to UVR is inflammation, such as erythema and edema, and DNA and mitochondrial damage caused by ROS [11]. The major ROS species involved in this process are superoxide radical anion (O2-), hydroxyl radical (OH), hydrogen peroxide (H2O2), and singlet oxygen species (O2) [12].”

Point 3: Lines: 97-98: "Keratinocytes build a physical water barrier by producing lipids and it also act as a defense system through regulating immune response [22]." This sentence is misleading since the main barrier towards TEWL resides in the stratum corneum (in particular the multilamellar lipid matrix of SC). Please clarify.

Response 3: We thank the reviewer for this comment regarding the how keratinocytes build water barrier and we have modified accordingly (page 3, lines 120-121). 

“Keratinocytes form a water barrier by means of the stratum corneum (SC), which is generated in the epidermal basal layer and the tight junctions, form a barrier in stratum granulosum [23].”

Point 4: Lines 98-101: "...many papers which investigated the effects of UVR in epidermis". What is the main effect of UVR on the stratum corneum?

Response 4: We would like to thank the reviewer’s comment regarding the main effect of UVR on the stratum corneum. We have included this information in the reviewed paper (page 4, 123-124).

              “The major SC damage caused by UV exposure includes rough and dehydrated texture, reduced desquamation and barrier function, and detrimental effects on cell cohesion [27].”

Point 5: Line 107: The abbreviation "EGFR" appears before its explanation at line 127.

Response 5: We would like apologize for this mistake. We have corrected this mistake in the revised article. (Page 4, line 129).

Point 6:  It would be helpful if the location of the signaling pathways presented in Figs. 2, 4, and 5 could be indicated. In other words, what regions of the skin organ, and in what cells, are they primarily occurring in.

Response 6: We would like to thank the reviewer for this comment. We have modified Figure 2, 4 and 5 according to the comment.

Point 7: Lines 234-236: "This can lead to release of inflammatory molecules from epidermal keratinocytes (IL-1, IL-3, IL-6, IL-8, GM-CSF, M-CSF, G-CSF, TGF-α, TGF- β, TNF- α and PDGF) and from dermal fibroblasts (MMP-1, MMP-2, MMP-3, MMP-9, MMP- 11, MMP-17 and MMP-236 27)." Please, either clarify the effects of UVR on other cell types (for example, in epidermis: melanocytes, Langerhans cells, Merkel cells; in dermis: vascular smooth muscle cell, macrophages, adipocytes, mast cells, Schwann cells, follicular stem cells) or state why the effect on these cells are not included.

Response 7: We appreciate the reviewer`s comment regarding this issue. We have explained this issue in the revised article. (page 2, lines 70-71, page 11, lines 299-300).

“In this review, our main focus is to elaborate effects of UVR on epidermal keratinocytes and dermal fibroblast in the skin aging process.”

“In skin, epidermal keratinocytes are the major source of cytokines production and dermal fibroblasts are the major source of MMPs and it has been demonstrated DNA damage leads to release of inflammatory molecules from epidermal keratinocytes (IL-1, IL-3, IL-6, IL-8, GM-CSF, M-CSF, G-CSF, TGF-α, TGF- β, TNF- α and PDGF) and from dermal fibroblasts (MMP-1, MMP-2, MMP-3, MMP-9, MMP- 11, MMP-17 and MMP-27) [48].”

Point 8: Abbreviations: It is recommended to include all abbreviations used in the text.

Response 8: We would like to appreciate reviewer’s comment and we have included all the abbreviations used in the text.

Response to Reviewer 3 Comments

The review by Ansary et al. was extremely thorough and informative.  They described how UVR can cause inflammatory responses in both the dermis and epidermis of the skin. 

We would like to thank the reviewer for the review and the important comments which have helped guide the revision of the manuscript. Our replies are as follows:

Point 1: In the abstract please omit etc. (it appears twice).

Response 1: We would like to thank this comment and we omit ‘etc.’ which appeared twice.

Point 2: In the introduction UVR is said to come from the sun and "artificial sources" could the authors elaborate on what those sources may be and why they are used?

Response 2: We thank the reviewer for this comment and we have included the information regarding the artificial sources of UVR in the text. (page 1, line 38, page 2, lines 39-40).

             “The major artificial sources of UVR are mercury vapor lamps, water-cooled lamps, and air-cooled lamps mainly used for diagnosis and treatment purposes.”

Point 3: Table 1, not just "Table" in the text and reference to it in the prose.

Response 3: We would like to thank the reviewer for this comment and we have modified Table 1 according to the suggestion (page 4, line 159, page 5, 197).

Point 4: Major comments:

I recommend a complete edit of the text as there are numerous grammatical mistakes which take away from the review.  The tone is often too conversational in some places (i.e. "on the other hand" is used often which seems to be a bit too informal).  Please edit the text.

Response 4: We would like to apologize for the poor English and grammatical mistakes in the review. We have taken reviewer’s comment seriously and we have edited the review by a native English speaker.

List of changes made in the revised manuscript-

  • The entire manuscript has been carefully checked, errors have been corrected, and explanations improved.
  • Figure 2, 4 and 5 have been modified.
  • Text was changed in the manuscript (page 1, lines 14, 38; page 2, lines 39-40, 43, 70-71, 74-75; page 3, lines 120-121; page 4, lines 123-124, 129, 130, 159; page 5, line 197; page 11, lines 299-300).
  • An additional 3 references have been cited (references #12, #23 and #27).

Reviewer 2 Report

This review summarizes skin aging caused by UVR and subsequent ROS mediated inflammation and associated circulatory inflammatory molecules and their inflammatory pathways. The topic is interesting and the review provides a relevant compilation of studies and important findings, in particular in the form of a comprehensive table. It is my opinion that the review could be published after minor revisions.

In general, the language could be improved (a linguistic review is recommended). Likewise,  the figures could be edited in a more compelling manner, including larger font size of the labels in the figures.

Some minor comments are given below.

The "table" should be referred to as Table 1.

Lines 56-57: ROS cause inflammation following acute exposure to UVR. Please include the major/most relevant ROS species in this instance.

Lines: 97-98: "Keratinocytes build a physical water barrier by producing lipids and it also act as a defense system through regulating immune response [22]." This sentence is misleading since the main barrier towards TEWL resides in the stratum corneum (in particular the multilamellar lipid matrix of SC). Please clarify.

Lines 98-101: "...many papers which investigated the effects of UVR in epidermis". What is the main effect of UVR on the stratum corneum?

Line 107: The abbreviation "EGFR" appears before its explanation at line 127.

It would be helpful if the location of the signaling pathways presented in Figs. 2, 4, and 5 could be indicated. In other words, what regions of the skin organ, and in what cells, are they primarily occurring in.

Lines 234-236: "This can lead to release of inflammatory molecules from epidermal keratinocytes (IL-1, IL-3, IL-6, IL-8, GM-CSF, M-CSF, G-CSF, TGF-α, TGF- β, TNF- α and PDGF) and from dermal fibroblasts (MMP-1, MMP-2, MMP-3, MMP-9, MMP- 11, MMP-17 and MMP-236 27)." Please, either clarify the effects of UVR on other cell types (for example, in epidermis: melanocytes, Langerhans cells, Merkel cells; in dermis: vascular smooth muscle cell, macrophages, adipocytes, mast cells, Schwann cells, follicular stem cells) or state why the effect on these cells are not included.

Abbreviations: It is recommended to include all abbreviations used in the text.

Author Response

Response to Reviewer 2 Comments

This review summarizes skin aging caused by UVR and subsequent ROS mediated inflammation and associated circulatory inflammatory molecules and their inflammatory pathways. The topic is interesting and the review provides a relevant compilation of studies and important findings, in particular in the form of a comprehensive table. It is my opinion that the review could be published after minor revisions.

In general, the language could be improved (a linguistic review is recommended). Likewise, the figures could be edited in a more compelling manner, including larger font size of the labels in the figures.

We thank the reviewer for the thoughtful review and important comments which have helped guide the revision of the manuscript. According to the reviewer’s suggestions we have completed a thorough English corrections and also increase the font size in figure legends. Our replies are as follows:

Point 1: The "table" should be referred to as Table 1.

Response 1: We thank the reviewer`s comment regarding how Table should be referred.  We have also modified accordingly (page 4, line 159, page 5, 197).

Point 2:  Lines 56-57: ROS cause inflammation following acute exposure to UVR. Please include the major/most relevant ROS species in this instance.

Response 2: We thank the reviewer for this comment regarding the major ROS species involved in UVR mediated inflammation. In the revised manuscript, we have added these information (page 2, line 74-75). 

“It is well understood that the acute response of skin to UVR is inflammation, such as erythema and edema, and DNA and mitochondrial damage caused by ROS [11]. The major ROS species involved in this process are superoxide radical anion (O2-), hydroxyl radical (OH), hydrogen peroxide (H2O2), and singlet oxygen species (O2) [12].”

Point 3: Lines: 97-98: "Keratinocytes build a physical water barrier by producing lipids and it also act as a defense system through regulating immune response [22]." This sentence is misleading since the main barrier towards TEWL resides in the stratum corneum (in particular the multilamellar lipid matrix of SC). Please clarify.

Response 3: We thank the reviewer for this comment regarding the how keratinocytes build water barrier and we have modified accordingly (page 3, lines 120-121). 

“Keratinocytes form a water barrier by means of the stratum corneum (SC), which is generated in the epidermal basal layer and the tight junctions, form a barrier in stratum granulosum [23].”

Point 4: Lines 98-101: "...many papers which investigated the effects of UVR in epidermis". What is the main effect of UVR on the stratum corneum?

Response 4: We would like to thank the reviewer’s comment regarding the main effect of UVR on the stratum corneum. We have included this information in the reviewed paper (page 4, 123-124).

              “The major SC damage caused by UV exposure includes rough and dehydrated texture, reduced desquamation and barrier function, and detrimental effects on cell cohesion [27].”

Point 5: Line 107: The abbreviation "EGFR" appears before its explanation at line 127.

Response 5: We would like apologize for this mistake. We have corrected this mistake in the revised article. (Page 4, line 129).

Point 6:  It would be helpful if the location of the signaling pathways presented in Figs. 2, 4, and 5 could be indicated. In other words, what regions of the skin organ, and in what cells, are they primarily occurring in.

Response 6: We would like to thank the reviewer for this comment. We have modified Figure 2, 4 and 5 according to the comment.

Point 7: Lines 234-236: "This can lead to release of inflammatory molecules from epidermal keratinocytes (IL-1, IL-3, IL-6, IL-8, GM-CSF, M-CSF, G-CSF, TGF-α, TGF- β, TNF- α and PDGF) and from dermal fibroblasts (MMP-1, MMP-2, MMP-3, MMP-9, MMP- 11, MMP-17 and MMP-236 27)." Please, either clarify the effects of UVR on other cell types (for example, in epidermis: melanocytes, Langerhans cells, Merkel cells; in dermis: vascular smooth muscle cell, macrophages, adipocytes, mast cells, Schwann cells, follicular stem cells) or state why the effect on these cells are not included.

Response 7: We appreciate the reviewer`s comment regarding this issue. We have explained this issue in the revised article. (page 2, lines 70-71, page 11, lines 299-300).

“In this review, our main focus is to elaborate effects of UVR on epidermal keratinocytes and dermal fibroblast in the skin aging process.”

“In skin, epidermal keratinocytes are the major source of cytokines production and dermal fibroblasts are the major source of MMPs and it has been demonstrated DNA damage leads to release of inflammatory molecules from epidermal keratinocytes (IL-1, IL-3, IL-6, IL-8, GM-CSF, M-CSF, G-CSF, TGF-α, TGF- β, TNF- α and PDGF) and from dermal fibroblasts (MMP-1, MMP-2, MMP-3, MMP-9, MMP- 11, MMP-17 and MMP-27) [48].”

Point 8: Abbreviations: It is recommended to include all abbreviations used in the text.

Response 8: We would like to appreciate reviewer’s comment and we have included all the abbreviations used in the text.

Reviewer 3 Report

The review by Ansary et al. was extremely thorough and informative.  They described how UVR can cause inflammatory responses in both the dermis and epidermis of the skin.  

Minor comments:

In the abstract please omit etc. (it appears twice).

In the introduction  UVR is said to come from the sun and "artificial sources" could the authors elaborate on what those sources may be and why they are used?

Table 1, not just "Table" in the text and reference to it in the prose.

Major comments:

I recommend a complete edit of the text as there are numerous grammatical mistakes which take away from the review.  The tone is often too conversational in some places (i.e. "on the other hand" is used often which seems to be a bit too informal).  Please edit the text.

Author Response

Response to Reviewer 3 Comments

The review by Ansary et al. was extremely thorough and informative.  They described how UVR can cause inflammatory responses in both the dermis and epidermis of the skin. 

We would like to thank the reviewer for the review and the important comments which have helped guide the revision of the manuscript. Our replies are as follows:

Point 1: In the abstract please omit etc. (it appears twice).

Response 1: We would like to thank this comment and we omit ‘etc.’ which appeared twice.

Point 2: In the introduction UVR is said to come from the sun and "artificial sources" could the authors elaborate on what those sources may be and why they are used?

Response 2: We thank the reviewer for this comment and we have included the information regarding the artificial sources of UVR in the text. (page 1, line 38, page 2, lines 39-40).

             “The major artificial sources of UVR are mercury vapor lamps, water-cooled lamps, and air-cooled lamps mainly used for diagnosis and treatment purposes.”

Point 3: Table 1, not just "Table" in the text and reference to it in the prose.

Response 3: We would like to thank the reviewer for this comment and we have modified Table 1 according to the suggestion (page 4, line 159, page 5, 197).

Point 4: Major comments:

I recommend a complete edit of the text as there are numerous grammatical mistakes which take away from the review.  The tone is often too conversational in some places (i.e. "on the other hand" is used often which seems to be a bit too informal).  Please edit the text.

Response 4: We would like to apologize for the poor English and grammatical mistakes in the review. We have taken reviewer’s comment seriously and we have edited the review by a native English speaker.

List of changes made in the revised manuscript-

  • The entire manuscript has been carefully checked, errors have been corrected, and explanations improved.
  • Figure 2, 4 and 5 have been modified.
  • Text was changed in the manuscript (page 1, lines 14, 38; page 2, lines 39-40, 43, 70-71, 74-75; page 3, lines 120-121; page 4, lines 123-124, 129, 130, 159; page 5, line 197; page 11, lines 299-300).
  • An additional 3 references have been cited (references #12, #23 and #27).
